

# X-band dual-polarization radar-based hydrometeor classification for Brazilian tropical precipitation systems

Jean-François Ribaud[1], Luiz A. T. Machado[1], and Thiago Biscaro[1]

[1]*National Institute of Space Research (INPE), Center for Weather Forecast and Climate Studies*
10       *(CPTEC), Rodovia Presidente Dutra, km 40, Cachoeira Paulista, SP, 12 630-000, Brazil*

*Correspondence to*: Jean-François Ribaud (jean-francois.ribaud@inpe.br)

**Abstract.**

The dominant hydrometeor types associated with Brazilian tropical precipitation systems are identified via research X-band dual-polarization radar deployed in the vicinity of the Manaus region (Amazonas) during both the GoAmazon2014/5 and ACRIDICON-CHUVA field experiments. The present study is based on an Agglomerative Hierarchical Clustering (AHC) approach that makes use of dual polarimetric radar observables (reflectivity at horizontal polarization $Z_H$, differential reflectivity $Z_{DR}$, specific

differential phase $K_{DP}$, and correlation coefficient $\rho_{HV}$) and temperature data inferred from sounding balloons. The sensitivity of the agglomerative clustering scheme for measuring the inter-cluster dissimilarities (linkage criterion) is evaluated through the wet season dataset. Both the weighted and Ward linkages exhibit better abilities to retrieve cloud microphysical species, whereas clustering outputs associated with the centroid linkage are poorly defined. The AHC method is then applied to investigate

the microphysical structure of both the wet and dry seasons. The stratiform regions are composed of five hydrometeor classes: drizzle, rain, wet snow, aggregates, and ice crystals, whereas convective echoes are generally associated with light rain, moderate rain, heavy rain, graupels, aggregates and ice crystals. The main discrepancy between the wet and dry seasons is the presence of both low- and high-density graupels within convective regions, whereas the rainy period exhibits only one type of graupel.

Finally, aggregate and ice crystal hydrometeors in the tropics are found to exhibit higher polarimetric values compared to those at mid-latitudes.

**Keywords:** hydrometeor identification, tropical microphysics, dual-polarization radar, clustering.





## 1. Introduction

The use of dual-polarization (DPOL) radars over several decades by national weather services as well as research laboratories has deeply changed the understanding and forecasting of many precipitation events around the world. By using a second orthogonal polarization, such weather radars enable inference of the size, shape, orientation, and phase state of different particles detected within the

sampled cloud. To date, the major advances that have been made as a result of DPOL radar sensitivities are mainly related to improvement in the distinction between meteorological and non-meteorological echoes, attenuation correction, quantitative rainfall estimation, and bulk hydrometeor classification (Bringi and Chandrasekar 2001; Bringi et al., 2007). By combining DPOL radar observables (generally, reflectivity at horizontal polarization, $Z_H$; differential reflectivity, $Z_{DR}$; specific differential phase, $K_{DP}$;

and correlation coefficient, $\rho_{HV}$) with some extra information such as temperature to locate the freezing level, the hydrometeor identification task has been the subject of many research studies. Indeed, potential benefits from this research topic are numerous such as the evaluation of microphysical parametrization in high-resolution numerical weather prediction models (e.g., Augros et al., 2016; Wolfensberger and Berne, 2018), investigation of relationships between microphysics and lightning

(e.g., Ribaud et al. 2016a), and improvement in weather nowcasting for high-impact meteorological events (hailstorms, flight assistance, road safety).

Three hydrometeor classification schemes have been developed since the emergence of DPOL radar in the 1980s: (i) supervised, (ii) unsupervised, and (iii) semi-supervised techniques (Figure 1).





i. The supervised method constitutes, by far, most of the literature and is subdivided into three different techniques: the boolean tree method, fuzzy logic and the Bayesian approach. Here, the supervised technique refers to a priori and arbitrarily identified hydrometeor types from which DPOL radar responses have been derived from either theoretical models or empirical knowledge. Polarimetric observations are then assigned to the most suitable hydrometeor types according to their similarities.

- Boolean method. This technique is the easiest way to identify dominant hydrometeor populations and has consequently been the first to be used. The algorithm relies on the beforehand definition of the ranges of DPOL radar-observable values for each hydrometeor type by the user. Then, a simple Boolean decision is applied to retrieve the dominant hydrometeor type (Seliga and Bringi, 1976; Hall et al, 1984; Bringi et al, 1986; Straka and Zrnić, 1993; Höller et al, 1994). This approach, nevertheless, does not take into account the fact that different hydrometeor types can be defined on the same range of values for the same polarimetric radar observable and, therefore, frequently leads to misclassification.

- Fuzzy logic technique (Mendel et al., 1995). This supervised algorithm type fixed the previous limitation by allowing a smooth transition of DPOL radar-observable ranges for all hydrometeor types. The originality of fuzzy logic is its ability to transform sets of nonlinear radar data into scalar outputs referring to different microphysical species. In this regard, each hydrometeor type distribution is characterized by a membership function coming from either T-matrix simulations (Mishchenko and Travis, 1998) or, less frequently, aircraft in situ measurements. The hydrometeor inference is finally the result of a combination of

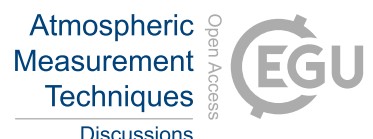

membership functions and a set of a priori rules defined by the user (Straka et al., 1996; Vivekanandan et al., 1999; Liu and Chandrasekar, 2000; Marzano et al, 2006; Park et al., 2009, Dolan and Rutledge, 2009; Al-Sakka et al., 2013; Thompson et al., 2014). This method is relatively simple to implement and computationally inexpensive. Few studies such as the Joint Polarization Experiment (Ryzhkov et al., 2005) for hail detection or even the recent use of a fuzzy logic algorithm as an operational tool for national weather services (Al-Sakka et al., 2013) have demonstrated the robustness of this hydrometeor classification algorithm type in singular environments.

- Bayesian approach. In this case, the hydrometeor identification task is expressed in a probabilistic form based on synthetic data derived from polarimetric radar simulation of different hydrometeor types (with each one being characterized by a centre and a covariance matrix). The final supervised hydrometeor inference is then performed by adapting the maximum a posteriori rule. Another interesting attribute of the Bayesian technique resides in the appealing possibility of retrieving the liquid water content associated with each hydrometeor type (Marzano et al., 2008; Marzano et al., 2010).

ii. More recently, Grazioli et al. (2015) or even Grazioli et al. (2017) proposed an innovative unsupervised approach to identifying the dominant hydrometeor distribution within precipitation events, where hydrometeor types are retrieved by gathering DPOL radar data observable similarities. Indeed, the unsupervised technique refers to a set of unlabelled data observations in which the goal is to group them into clusters sharing similar properties based on innate structures of the data (variance, distribution, etc.) and without using a priori knowledge. To

achieve this goal, the authors used an agglomerative hierarchical clustering technique together with a spatial constraint on the consistency of the classification (homogeneity). This data-driven approach mainly avoids the numerical-scattering simulations used in fuzzy logic, which are well-designed for the liquid phase but questionable for ice-phase microphysics. Finally, interpretation of the clusters (labelling) is done manually.

iii. Although initially mentioned by Liu and Chandrasekar (2000), the first complete study based on a semi-supervised approach was done by Bechini and Chandrasekar (2015), recently followed by the works of Wen et al. (2015), Wen et al. (2016) and Besic et al. (2016). This technique combines the advantages of the fuzzy logic and clustering methods. The algorithm initially begins with a fuzzy logic classification, which is then adjusted by a K-means clustering method that iteratively allows for rectifying the initial membership function of each hydrometeor type according to the observed DPOL radar measurements. In addition, constraints such as temperature limits and/or spatial distribution can be implemented in this self-adapting methodology.

Overall, these Hydrometeor Classification Algorithms (HCAs) still require in situ aircraft validations (especially within convective cores) that are problematic due to their cost and, obviously, the dangerousness of obtaining such measurements. Only a few studies have had the opportunity to use limited aircraft measurements and generally compared a few isolated in situ images with HCA outputs (Aydin et al., 1986; El-Magd et al., 2000; Cazenave et al., 2016; Ribaud et al., 2016b). Another limitation of these studies using methods such as the fuzzy logic approach is the dependency of their



validity, since they are generally both wavelength- and climatically radar-dependent. Although T-matrix

simulations for a radar wavelength have been theoretically demonstrated, each final algorithm is then

tuned by giving weights to each DPOL radar observable to allow them to fit as closely as possible with

local ground observations. Finally, one can also see that the related hydrometeor identification literature

is mainly concerned with the middle latitudes. Indeed, the methods were initially developed for S-band

radar before being adapted to both C- and X-band radars, and research studies have largely been done in

North America, Europe, and Oceania.


The present study aims to develop the first HCA for Brazilian tropical precipitation systems via an

X-band dual-polarization radar used in both the GoAmazon2014/5 and ACRIDICON-CHUVA field

experiments (Martin et al., 2016; Wendisch et al.,2016; Martin et al., 2017; Machado et al., 2018).

Although the area constitutes an intriguing location with both a high amount of rain and complex

aerosol-cloud interaction (e.g., Cecchini et al., 2017; Machado et al., 2018), there are almost no

references for hydrometeor classification over tropical land, especially for the Amazon region. In this

regard, the studies by Dolan et al. (2013) and Cazenave et al. (2016) took place in singular locations

(Darwin, Australia, and Niamey, Niger, respectively). Both of these studies used a supervised fuzzy

logic approach to retrieve the hydrometeor distribution within precipitation events with a C- and

adapted X-band scheme, respectively. As aforementioned, fuzzy logic algorithms use weights to

constrain the final identification. Another issue that might be related to hydrometeor identification tasks

is the use of the melting layer as a parameter to detect liquid-ice delineation. However, liquid water

above the melting layer within the convective tower of tropical systems is not unusual (Cecchini et al.,



2017; Jakel et al., 2017). For instance, Cecchini et al. (2017) retrieved liquid water at as low as -18 °C

within polluted tropical convective clouds. Classification using cluster analysis might thus tackle this

issue by allowing the use of natural (non-imposed) classes of ice-water species. For all these reasons,

the present paper deals with the first unsupervised clustering method based on X-band DPOL radar

measurements in the Brazilian tropical region. Three main questions are addressed in this paper: (1)

What is the sensitivity of the clustering algorithm to the different linkage methods, and how can one

improve the liquid-solid delineation? (2) What are the hydrometeor classification output characteristics

for both wet and dry tropical seasons in Amazonas? And (3) what are the microphysical distribution

differences within tropical convective and stratiform cloud systems between the wet and dry seasons?

The article is organized as follows: section 2 provides a brief description of the radar dataset,

while section 3 presents the AHC method. The sensitivity of the AHC to the linkage methods together

with a potential temperature improvement is assessed and discussed in section 4. The hydrometeor

identification for Brazilian tropical system events is presented in terms of wet-dry seasons and

stratiform-convective regions in section 5, while a discussion of hydrometeor distribution comparisons

is presented in section 6.

## 2. Datasets and processing

The data used in this study are mainly based on DPOL radar data observations collected during

both the GoAmazon2014/5 and ACRIDICON-CHUVA experiments that took place around the city of

Manaus in the Amazonas state of Brazil (Figure 2). Both of these research experiments aimed to

investigate the complex mechanisms at play within tropical weather through intriguing interactions



between human activities and the neighbouring tropical forested region. In this regard, the present study

considers the wet and dry seasons as corresponding to the intensive operating periods (IOPs) of the

GoAmazon2014/5 field experiment (Martins et al., 2016), which were from 1 Feb – 31 Mar 2014 (wet

season: 59 days) and 15 Aug – 12 Oct 2014 (dry season: 60 days).

Among all the instruments deployed, a Selex-Gematronik X-band DPOL radar was located in the

city of Manacapuru in 2014 to complete the radar coverage from the Manaus Doppler radar, as well as

to provide more microphysical details about the South American monsoon meteorological systems

(Oliveira et al., 2016). The X-band DPOL radar was operated at 9.345 GHz with a 1.3° beam width at -3

dB and in simultaneous transmission and reception (STAR) mode (Schneebeli et al., 2012; and Table 1).

The latter characteristic allows the reflectivity at horizontal polarization $Z_H$, differential reflectivity $Z_{DR}$,

differential phase $\Phi_{DP}$, and correlation coefficient $\rho_{HV}$ to be obtained. The scanning strategy was

designed to complete an entire volume scan in 10 minutes by combining 15 different plan position

indicators (PPIs) ranging from 0.5° to 30°, as well as two range height indicators (RHIs) towards

randomly different directions.

The raw radar dataset has been processed beforehand to be used for the hydrometeor identification

task. In this regard, a four-step process has been applied to the DPOL radar dataset which consists of i)

calibration of $Z_{DR}$ (offset corrected by vertically pointing scans), ii) identification of meteorological and

non-meteorological echoes, iii) $\Phi_{DP}$ filtering and estimation of the derivative specific differential phase

$K_{DP}$ (Hubbert and Bringi, 1995), and iv) attenuation correction applied to both $Z_H$ and $Z_{DR}$ based on the

ZPHI method proposed by Testud et al. (2000). Note that the dataset has been restricted to precipitation

events wherein the radome of the X-band DPOL radar was dry in order to remove any additional

attenuation (Bechini et al, 2010). In addition to these considerations, a signal-to-noise ratio of SNR ≥ +10 dB, as well as a reduced radar coverage ranging from 5 to 60 km have been considered for this study to mitigate potential remaining errors. The last processing step relies on the separation of stratiform and convective radar echoes. The methodology used in the present paper is the same as that

used by Steiner et al. (1995) and has been applied from a horizontal reflectivity field at a constant altitude plan position indicator (CAPPI) generated at 3 km height (T > 0 °C).

The present study also deals with external temperature information coming from soundings launched near the X-band radar (downwind of Manaus) at 00, 06, 12, 15, and 18 UTC, respectively. The sounding with the closest time to the radar measurements has been considered to derive the temperature

profile associated with both PPIs and RHIs.

## 3. Unsupervised Agglomerative Hierarchical Clustering

The present hydrometeor classification algorithm is an unsupervised AHC method that aims to partition a set of n observations into N different clusters. This technique works as an iterative "bottom-

up" method where each observation starts in its own cluster and pairs of clusters are aggregated step by step until there is one final cluster, which comprises the entire dataset. Each cluster is composed of a group of observations sharing more similar characteristics than the observations belonging to the other clusters. Here, there is no a priori information concerning the shape and size of each cluster or the final optimized number of clusters. A posteriori analysis is then performed through the final iterations to

retrieve the optimal clustering partition and respective labels.



Since associated background already exists, the reader is especially referred to Ward (1963) and Jain et al. (2000) for detailed mathematical reviews of the technique. Additionally, the present clustering framework is mainly based on the methodology developed by Grazioli et al. (2015 – section 4 and Figure 2), hereafter referred to as GR15, and only relevant and important information will be addressed

hereafter to avoid being redundant. The main steps of the present AHC can be summarized as follows:

- Vectorized objects of radar observations are defined for each valid radar resolution volume as

$$x = \{Z_H, Z_{DR}, K_{DP}, \rho_{HV}, \Delta z\},$$

- where $\Delta z$ is the difference between the radar resolution height and the altitude of the isotherm at 0 °C, deduced from sounding balloons.

- Since scales of radar polarimetric variables differ by orders of magnitude, data normalization is applied to concatenate all the observations into a [0;1] common space. The first four components of each object are based on the minimum-maximum boundaries rule. The temperature information is redistributed by applying a soft sigmoid transformation that allows setting a value of zero (one) for altitudes below (over) the bright band. Here, the thickness of the bright band

over the whole GoAmazon2014/5 – ACRIDICON-CHUVA database has been manually estimated and set up to spread over a layer of ± 700 m. To obtain the maximum degrees of freedom in the initial dataset coming from the DPOL radar measurements, here, the influence of the temperature information is mitigated by distributing its values into a [0;0.5] range space.

- Although the radar data are now suitable for clustering, the choice of two criteria still remains.

At each iteration of the AHC method, similarities/dissimilarities must be evaluated to determine which clusters merge. In this regard, the Euclidean metric is considered to calculate the distance



between different single objects. The generalization of this distance metric to an ensemble of objects is called the merging linkage rule. Various methods exist to evaluate inter-dissimilarities such as single (nearest neighbour), complete (farthest neighbour), averaged, weighted, centroid, or even Ward (variance minimization) linkages (see Müllner, 2011). Herein, we consider the weighted, centroid and Ward linkage rules (see section 4.a).

- Running such a clustering method over the whole dataset is computationally very expensive. To tackle this problem, a subset of approximately 25 000 initial observations is randomly chosen through the whole precipitation events database. The clustering method is initially applied to the subset and then extended to the whole dataset by using the nearest cluster rule at each iteration.

- One of the major novelties proposed by GR15 relies on the implementation of a spatial constraint that aims to check the homogeneity of the clustering distribution at each iteration. More precisely, one assumes that a smooth, horizontal transition exists between the resulting hydrometeor field outputs. Therefore, a spatial smoothness index is calculated at the end of each iteration step and individual object by checking the four closest geographical radar gates. In the very same way as that used in GR15, results are summarized into a confusion matrix, from which several spatial indexes can be extracted to analyse the individual and global spatial smoothness of a partition.

- The merging of two clusters is realized by identifying the cluster which presents the lowest spatial similarities among all clusters. Objects belonging to this spatially poor cluster are then



constrained to be redistributed through the other existing clusters according to the linkage method chosen. This final step allows decreasing the total number of clusters by one.

- If the iteration process does not reach a single and unique cluster, the iteration loop then restarts at the initial PPIs classification and goes through the evaluation of spatial homogeneity.

- Finally, an analysis of the variance explained has been implemented to evaluate the consistency of the clustering classification outputs. This quality metric allows definition of the theoretically appropriate number of clusters by analysing the ratio between the internal and external variance of each cluster at each step of the iteration. The main idea here is to find the optimal cluster distribution beyond which considering one more cluster is not meaningful.


## 4. Methodology discussions

### a) Linkage rule sensitivity

According to the setup described in section 3, different linkage rules have been tested through the special wet season observation period (February to March) of 2014. To perform this sensitivity test,
three different linkage rules have been considered here: (i) weighted, (ii) centroid, and (iii) Ward (see Table 2 for their respective formulas). Since the clustering method randomly picks observations within the whole wet season period, a set of numerous runs for each linkage method have been performed to extract, as much as possible, the most representative behaviour of each one. The general common setup is composed of a subset of 25 000 observations randomly picked through more than 50 precipitation
days. The temperature information is based on radiosounding observations and is dispatched in a [0;0.5] interval to place twice as much importance on the initial DPOL radar observations. The number of

clusters reached in the first step of the AHC method is set at 50 (far enough from the final partition and not too computationally expensive). Finally, the clustering method has been conducted separately on stratiform and convective regions.


In this respect, Figure 3 presents the evolution of the variance explained (the ratio between the internal and external variance) for the three different linkage rules as a function of the number of clusters considered, together with their associated precipitation regimes (stratiform or convective). Overall, the three methods exhibit an "elbow" curvature with an optimal number of clusters ranging from approximately 5 to 8 (orange background on Figure 3). One can see that from 2 to 5 clusters, the variances explained sharply increases, meaning that each added cluster within this interval contributes significantly to retrieving the most adequate cluster partition. From 5 to 8 clusters, the increase starts to slow down, indicating that considering a greater number of clusters is not meaningful. In this regard, the best "compromise" seems to be the weighted and/or Ward linkage method for both stratiform and convective regions. Indeed, these methods have the highest scores, with approximately 99 % reached within the 5-8 clusters interval.

Due to the inherent complexity of representing all the potential combinations, manual analysis and selection have been performed beforehand to find the optimal number of clusters between the stratiform and convective regions. The results from this partitioning are presented through one stratiform and one convective RHI (Figures 4 and 5).

In addition, fuzzy logic information has been implemented to make comparisons with cluster outputs. The fuzzy logic scheme is mainly based on the X-band algorithm of Dolan and Rutledge (2009),

hereafter referred to as DR09, and has been slightly enriched for the wet snow and melting hail hydrometeor types by Besic et al (2016) through scattering simulations and a temperature membership function (Besic et al, 2016 – Appendix A). Finally, the adapted fuzzy logic allows discrimination between nine hydrometeor types: light rain (LR), rain (RN), melting hail (MH), wet snow (WS), aggregates (AG), low-density graupel (LDG), high-density graupel (HDG), vertically aligned ice (VI),

and ice crystals (IC).

Figure 4 shows a stratiform system exhibiting a well-defined bright band signature from polarimetric observations that occurred on the shores of the Amazon river on 21 February 2014. Overall, the centroid linkage method does not reproduce the event well, and the final representation is microphysically poor

(Figure 4-f). Indeed, this linkage rule simply divides the cloud into three homogeneous regions (T > 0 °C, T ~ 0 °C, and T < 0 °C). Additionally, the centroid linkage fails to identify a clear bright band region (Figure 4f, clusters 2S and 3S). On the other hand, the weighted and Ward linkage methods are very close to the fuzzy logic output descriptions (Figure 4e-g-h). They both exhibit two kinds of rain, and a bright band region sits on top of what appears to be an aggregates-ice crystals mixture. The main

discrepancy here concerns the representation of the rain structure. The Ward linkage rule retrieves two more distinct liquid species (as does fuzzy logic), whereas the weighted linkage method exhibits a smoother rainy region.



Figure 5 presents a decaying convective cell that occurred on 02 February 2014 at 13:57 UTC (0-7 km

from the radar: stratiform region, 7-40 km from the radar: convective region). As is the case for the

stratiform RHI in Figure 4, the centroid linkage rule fails to retrieve a detailed microphysical structure

and only presents very homogeneous liquid and solid regions. Once again, both the weighted and the

Ward linkage rule stand out and display a more realistic hydrometeor description of the convective

cloud in comparison to the DPOL radar observations and the fuzzy logic outputs (Figure 5 a-b-c-d-e-g-

h). Although they both present three clusters for T > 0 °C, the weighted linkage rule puts more emphasis

on the convective region located ~ 20-30 km from the radar than does the Ward linkage (Figure 5-e,

cluster 6C vs. Figure 5-g, cluster 11C). The representation of the solid region (T < 0 °C) is almost the

same, except for in the aggregates region (Figure 5h), which seems to be smaller for the weighted

linkage rule (Figure 5e cluster 8C) than for the Ward method (Figure 5g cluster 10C). Another

discrepancy between the weighted and Ward linkages concerns the layer around the isotherm at 0 °C.

Although Figure 5 does not exhibit any bright band region, the Ward linkage rule does exhibit one due

to the temperature input (Figure 5g cluster 12C), whereas the weighted rule does not. The bright band

region is known to be well-defined for stratiform regimes but quasi-undetectable (if detectable at all) for

convective areas (Leary and Houze, 1978; Smyth and Illingworth, 1998; Matrosov et al., 2007).

Throughout the present paper, one will thus consider only a bright band cluster for the stratiform

regions, whereas convective areas will be lacking one.

Overall, Figures 3, 4, and 5 have shown that the centroid linkage method is inappropriate for the present

task, whereas both weighted and Ward linkage rules are able to retrieve a detailed microphysical



structure within the sample cloud. Based on the present description and our personal analysis over the

whole dataset, we chose to keep working with the weighted linkage rule throughout the remainder of the

paper.

### b) Potential improvement around isotherm 0 °C

High amounts of liquid water a few kilometres above the isotherm at 0 °C are not rare within the core of

convective tropical cells. Sometimes, super-cooled liquid drops can be maintained and even moved

upward within the melting layer, thus occasionally giving distinctive column-shaped polarimetric

signatures for $Z_{DR}/K_{DP}$ (e.g., Kumjian and Ryzhkov, 2008). A simple liquid-solid delineation based only

on the temperature profile is therefore unsuitable.

Figure 6 presents an adaptive solution to tackle this issue based on the clustering outputs of the

weighted linkage rule. The solution proposed here relies on a posteriori analysis of the clustering

outputs associated with the convective regions. First, one proceeds to identify the convective core under

the isotherm at 0ºC (here, cluster 6C). Then, all radar observations within the solid region are assigned

by calculating their distance from the 6C cluster centroid without applying any temperature constraint

(objects are thus defined only by the first four radar components). If the distance is smaller than D<0.25

and there is no discontinuity throughout the liquid-solid delineation, then the solid identification is

switched to liquid (cluster 6C). Note that the distance D has been empirically chosen for the present

radar observations and could consequently be adjusted by exploring more convective days. Overall,

with this simple hypothesis, one can see the potential of a such method (Figure 6b). The liquid cluster

can thus reach 8 km in the core of the convection at 25 km from the radar, which matches well with the

convective tower (>35 dBZ) visible in Figure 5a. Around this convective core, the enhancement allows

raising raindrops by about one kilometre upward in the 0ºC isotherm, restraining cluster 6C at ~ 5 km.

In comparison to a simple binary delineation such as that used for the fuzzy logic outputs (Figure 6a),

the focus on radar observables in a second phase is then promising.


## 5. Wet and dry season dominant hydrometeor classifications

This section aims to interpret and label each cluster retrieved through both the wet and dry seasons

over the Manaus region by using the AHC method setup described in section 3. As the use of

classification allowing liquid water above the melting layer of convective towers needs further

validation, a standard classification is used to classify and analyse the wet and dry hydrometeors using

the temperature parameter.

### a) Wet season clustering outputs

The distributions of $Z_H$, $Z_{DR}$, $K_{DP}$, $\rho_{HV}$, and $\Delta z$ for each cluster from the stratiform and convective

clouds of the wet season together with their probability densities are presented in the violin plot in

Figure 7 and Figure 8, respectively. The contingency table between the stratiform (convective)

clustering outputs and the nine microphysical species retrieved by the DR09 adapted fuzzy logic

algorithm is shown in Table 3 (Table 4). The complete wet season cluster centroids are given in

Appendix A.1.



### 1) Stratiform region

Cluster 1S is only defined for negative temperatures and is associated with high $\rho_{HV}$ and low $Z_H$,

$Z_{DR}$ and $K_{DP}$ values (Figures 4e and 7). One can see from contingency Table 3 that the cluster 1S

repartition is mostly associated with aggregates (~ 33 %) and ice crystals (~ 12 %) for high altitudes.

Although the horizontal and differential reflectivity values are slightly higher than those for the DR09

T-matrix microphysical outputs and polarimetric characteristics retrieved by GR15, one can make the

assumption that the cluster 1S behaviour stands for ice crystals. On the other hand, cluster 2S is closer

to the DR09 T-matrix aggregates microphysical features. This cluster is characterized by a mean

horizontal (differential) reflectivity of ~ 27 dBZ (~ 1.3 dB), a low specific differential phase (~ 0.27

degree/km) and a high coefficient of correlation (0.97). Overall, the polarimetric signatures of cluster 2S

are mostly divided into the associated wet and dry snow (aggregates) from the microphysical categories

of fuzzy logic (Table 3). Figure 4e allows discrimination between these categories, and one can consider

that cluster 2S is here associated with aggregates. Once again, its polarimetric signatures are slightly

higher than the DR09 T-matrix values or even the GR15 aggregates clustering output. One explication

behind these distributions being slightly shifted to higher values can be the relative humidity, which is

higher in the tropics than at higher latitudes. The growth of ice crystals/aggregates by vapor diffusion

within this cloud region (Houze, 1997) may lead to bigger solid particles (higher $Z_H$ and $Z_{DR}$ values).

The bright band region is well-represented here by cluster 4S. Indeed, its global distribution spreads

only at the altitude of the isotherm at 0 °C and exhibits high $Z_H$ and $Z_{DR}$ values, as well as low $K_{DP}$ and

$\rho_{HV}$ values. Finally, clusters 3S and 5S present rain characteristics since more than 90 % of these clusters

are in agreement with the drizzle and rain fuzzy logic types from DR09. Although the two clusters have

the same behaviours, cluster 3S is characterized by polarimetric signatures higher than those in cluster

5S, except for the coefficient of correlation (0.97 vs. 0.99, respectively). In this regard, one can consider

that cluster 3S represents the rain microphysical species, whereas cluster 5S is related to drizzle

characteristics.

### 2) Convective region

Overall, one can see from Figures 5 and 8 that the convective regions of the wet season are composed of

three types of hydrometeors for both positive (clusters 6C-10C-11C) and negative temperatures (clusters

7C, 8C and 9C).

Hail precipitation in the Amazonas region is rare, and as expected, no clusters represent melting hail

characteristics, as in Ryzhkov et al. (2013) or Besic et al. (2016) (Table 4). Therefore, clusters 6C, 10C,

and 11C can be associated with three distinct rainfall precipitation regimes. In this regard, cluster 10C

presents the same light rain characteristics as both DR09 and GR15. The cluster is characterized by $Z_H$

($Z_{DR}$) values approximately 13 dBZ (0.68 dB), and a $K_{DP}$ (0.14 degree/km) that is in high agreement

with the drizzle hydrometeor type from the adapted fuzzy logic (~ 97 %, Table 4). According to this

description, one can attribute cluster 11C to the light rain precipitation type. The two remaining liquid

clusters are associated with moderate and heavy rainfall types with almost the same polarimetric

signatures as those given in GR15. Indeed, cluster 6C presents higher $Z_H$ (44 vs. 31 dBZ), $Z_{DR}$ (2.1 vs

1.4 dB), and $K_{DP}$ (1.9 vs 0.8 degree/km) mean values than those for cluster 11C. In this regard, one can

link cluster 6C to heavy rainfall and cluster 11C to moderate rainfall.

Concerning negative temperatures, cluster 9C stands out by being spread at the highest altitudes (Figure

8-e). This cluster is defined by low $Z_H$, $Z_{DR}$, and $K_{DP}$ values together with a moderate $\rho_{HV}$ (~ 0.97). One

can note that cluster 9C is close to the ice crystals/small aggregates retrieved by GR15 and is also the

only cluster related to the T-matrix ice crystals species from DR09 (Table 4). Within the decaying

convective cell presented in Figure 5, one can observe that cluster 7C is associated with the low-density

graupel characteristics proposed by DR09 and exhibits $Z_H$ ($Z_{DR}$) values approximately 36 dBZ (0.8 dB).

In addition, cluster 7C is mainly classified (~ 69 %) as low-density graupel (Table 4). Finally, the last

cluster, 8C, is surrounded by ice crystals and presents polarimetric signatures lower than those for

cluster 7C. Although it is defined by higher values than those given by DR09 and GR15, one can

associate cluster 8C with the aggregate microphysical species. Indeed, contingency Table 4 shows that

45 % of the cluster 8C points are in agreement with this hydrometeor type.


**b) Dry season clustering outputs**

As for the previous section, the clustering outputs retrieved by the AHC method and the weighted

linkage rule are identified and associated with their corresponding microphysical species through the

dry tropical season. The corresponding cluster centroids are detailed in Appendix A.2.


**1) Stratiform region**

Figure 9 shows the clustering classification outputs extracted from an RHI presenting a melting layer region within a stratiform event that occurred on 08 September 2014 in the region of Manaus. Overall, the clustering outputs are close to the hydrometeor distribution retrieved by the adapted DR09 fuzzy

logic. Clusters 1S-2S retrieved for positive temperatures appear well located in terms of polarimetric signatures and fuzzy logic outputs. One can see that the melting layer region is clearly characterized by cluster 4S, whereas for negative temperatures, clusters 3S-5S show patterns close to the fuzzy logic outputs.

The violin plots in Figure 10 and contingency Table 5 allow discrimination and labelling of these

clusters. For DR09 classification, clusters 1S and 2S exhibit rainfall signatures. Cluster 2S is in agreement with the fuzzy logic drizzle category (~ 92 %), whereas cluster 1S is divided into the drizzle (~ 76 %) and rain (~ 22 %) microphysical species. Between these two clusters, one can observe that cluster 1S contains the highest $Z_H$, $Z_{DR}$ and $K_{DP}$ values, and one can consequently label it as a rainfall type. Cluster 2S is, however, associated with the drizzle/light rain category according to the polarimetric

radar signatures (GR15).

The liquid-solid delineation is represented here by cluster 4S. It presents a low $\rho_{HV}$ (~ 0.93) and a large $Z_H$ distribution around ~ 30 dBZ and is almost only defined for altitudes close to the 0°C isotherm. In addition, contingency Table 5 matches well with this hydrometeor association.

For the negative temperatures, the clustering outputs exhibit two clusters, 3S-5S. The first is located

within the edge region of the cloud, whereas cluster 5S is distributed at lower altitudes and is closer to particles of greater densities (Figure 10). Cluster 5S is in ~ 70 % agreement with the aggregate fuzzy logic outputs (Table 5), and its polarimetric signatures are close to those of GR15 and T-matrix

simulations from DR09. One can then define cluster 5S as the aggregate microphysical species. Finally,

ice crystals/small aggregates are represented through cluster 3S, which is defined by low $Z_H$, $Z_{DR}$, and

$K_{DP}$ values and a high $\rho_{HV}$.

### 2) Convective region

Figure 11 shows an RHI of a convective system that occurred in the late afternoon on 06 October 2014

in the region of Manaus. Overall, this RHI shows a convective cell (at 24-50 km from the radar)

together with its relative stratiform region (0-23 km). Note that the abrupt transition from the convective

and stratiform classification areas (Figure 5-6-11) is inherent to the Steiner et al. (1995) algorithm. In

terms of microphysical distribution, there should be some consistency between the two cloud types. The

implementation of continuity analysis may prevent the latter artefacts. The convective cell is

characterized by $Z_H$ values up to 25 dBZ at 14 km, and the cloud top exceeds 16 km. According to the

fuzzy logic outputs (Figure 11-f), the cell exhibits mostly rainfall precipitation for positive

temperatures. The corresponding cluster outputs retrieve the same signatures, dividing the rain pattern

into three different clusters: 6C, 7C, and 12C. Once again, the fuzzy logic collocates a bright band

around the isotherm at 0ºC, whereas neither polarimetric signatures nor clustering outputs exhibit a

bright band. For negative temperatures, the AHC method retrieves four clusters (8C, 9C, 10C and 11C)

as the fuzzy logic outputs.

The violin plots in Figure 12 and contingency Table 6 allow discrimination and labelling of these

clusters. For the convective regions observed during the wet season, hail precipitation is rare in the



Amazonas. Contingency Table 6 is also in agreement with this description, since none of the clustering

outputs exceed 2 %. Therefore, one can attribute clusters 6C, 7C, and 12C to three different rainfall

precipitation regimes, ranking the cluster positions as follows: 12C presents weaker $Z_H$, $Z_{DR}$, and $K_{DP}$

values than does cluster 7C, which presents lower values than does cluster 6C (Figure 12). In addition,

one can see from contingency Table 6 that all three are in very high agreement with the drizzle and rain

microphysical species. Based on the aforementioned description together with Figure 11 analysis, one

can attribute cluster 12C to light rainfall, cluster 7C to moderate rainfall and, finally, cluster 6C to the

heavy rainfall type.

Concerning all clusters spreading at negative temperatures, cluster 11C matches well with the high-

density graupel category defined by DR09 such as "graupel growing in regions of large supercooled

water contents, melting graupel, and freezing of supercooled rain". Based on contingency Table 6, this

cluster is mainly associated with wet snow and slightly with the low-density graupel microphysical

specie. Nevertheless, one can see that the $\rho_{HV}$ distribution is pretty low (~ 0.94) and could also be the

signature of wet graupel (due to melting or wet growth) or a mixture of graupel and hail, as suggested

by Straka et al (2000) and Kumjian et al (2008). This cloud region is surrounded by low-density

graupel, characterized by cluster 9C (Figures 11-12). This hydrometeor type shows 60 % agreement

with this microphysical type within contingency Table 6 and is close to the DR09 T-matrix outputs.

Cluster 10C shares more than 50 % with the aggregates type and 30 % with the low-density graupel

type, whereas cluster 8C is associated in general with ice crystals and aggregates types (Table 6). With

Figures 11-12 and the aforementioned description, one can analyse cluster 9C as low-density graupel,

cluster 10C as aggregates, and, finally, cluster 8C as ice crystals.




## 6) Discussion

### a) Impact of the clustering method and location

The present results allow making a brief comparison between the classical supervised fuzzy logic

technique commonly used in the literature and the unsupervised AHC method. In opposition to the rigid

structure of a fuzzy logic algorithm, the flexibility of the clustering approach allows better identification

of the bright band region. Indeed, the liquid-solid delineation around the 0 °C isotherm is better

captured and distinguished by the AHC method, which preferentially follows the polarimetric signatures

instead of the stratified temperature region. Additionally, one can see the ability of the AHC method to

fully exploit the high sensitivity of the X-band radar frequency to distinguish between three different

(light, moderate, and heavy) rainfall regimes such as in GR15. This enhancement allows, for instance,

putting more emphasis onto severe convective precipitation cells and may open new perspectives for

nowcasting issues.

Note that the present clustering method has been distinctly subdivided into stratiform and convective

regions. Although they are characterized by different thermodynamic structures (Houze, 1997), the

stratiform and convective regions may be related in terms of microphysical distributions, such as ice

particles which might be ejected from the top of an active convective cell into the upper part of the

stratiform region. This microphysical inconsistency may be prevented by the implementation of an a

posteriori continuity analysis.

The location of the present study also offers the possibility to discuss mid-latitude and tropical

microphysical differences. As described in section 5, the dominant tropical hydrometeor classification



overlaps some mid-latitude microphysical species definitions. For instance, one can see that both the

aggregate and ice crystal microphysical species are skewed to higher horizontal (differential)

reflectivity, regardless of the season and region (stratiform/convective) considered. These discrepancies

might be attributed either to an inaccurate attenuation correction (overestimations due to the ZPHI

method) or inherent tropical characteristics involved within microphysical ice growth. Indeed, tropical

atmospheric characteristics present higher tropospheric humidity profiles together with higher incident

solar radiation that could play an important role in comparison to mid-latitudes.

### b) Wet-Dry season differences

The investigation of some Amazonian wet-dry season differences has already been explored by a few

studies. For instance, Machado et al. (2018) noted that during both the GoAmazon2014/5 and

ACRIDICON-CHUVA field campaigns, the wet season overall mean cumulative rain was four time as

much as that during the dry season. However, though characterized by a low amount of total rainfall, the

dry season presents the higher rainfall rate (Dolan et al, 2013; Machado et al, 2018). According to

Machado et al (2018), these discrepancies can partly be explained by the fact that the dry season

presents higher convective available potential energy (CAPE) and lower cloud cover than those during

the wet season. Another study conducted by Giangrande et al (2017) also examined the wet-dry season

differences through convective clouds. The authors showed that warm clouds exhibit larger cloud

droplets and that the stratiform region during the wet season is much more developed than that during

the dry season (due to surrounding monsoon ambient characteristics).



All these differences are expected to contribute to the wet-dry season differences. Here, one can address for the first time these discrepancies through the dominant microphysical patterns in terms of stratiform/convection precipitation regimes associated with the Central Amazonas (Manaus region). Based on this new hydrometeor classification adapted to the tropical region, this section explores the differences among the clouds related to these two seasons.

### 1) Stratiform region

Figure 13 presents a comparison of pairs of stratiform hydrometeor types between the wet and dry seasons. For positive temperatures, both the drizzle and rain microphysical species present higher $Z_H$ and lower $Z_{DR}$ values during the dry season than during the wet season. These polarimetric signatures are generally attributed to the evaporation and breakup processes that tend to reduce the particle diameters (Kumjian and Ryzhkov 2010; Penide et al, 2013). However, this pattern normally refers to the dry season, which presents a more favourable environment. The separation between the drizzle/light rain and the rain microphysical species is defined for a rainfall rate of approximately 2.5 mm/h (American Meteorological Society, 2018). The classical Marshall-Palmer Z-R relationship allows estimation of the rainfall rate for stratiform precipitation. In this regard, the wet rain microphysical species is characterized, on average, by a rainfall rate of 1.84 mm/h, whereas the rate is up to 3 mm/h during the dry season. The general wet rain microphysical species distribution thus still contains drizzle/light rain observations. This puzzling rain partitioning might be due to the different cloud cover patterns associated with stratiform echoes during the two seasons. As noted by Machado et al (2018), stratiform cloud cover related to the rainy season is more associated with a monsoon cloud regime than



during the remaining season. While the dry season stratiform regime is directly the result of the rain convective cells, the wet stratiform cover may also refer to large ambient unrelated residual precipitation far outside the original convective cloud.

Overall, the melting layer, which is represented here through the wet snow microphysical species, is consistent with the results of previous studies (Durden et al, 1997; Giangrande et al, 2008; Heymsfield et al, 2015; Wolfensberger et al, 2015). The vertically restricted layer of wet snow presents the most widespread distribution of $Z_H$, $Z_{DR}$, $K_{DP}$ and $\rho_{HV}$ of all the retrieved microphysical species and for both seasons. One can see that the wet season distribution differs from the dry season, as its distribution is

more associated with lower (higher) $Z_H$ ($Z_{DR}$) values. The main discrepancy here is related to the $Z_{DR}$ distribution, which has stronger values during the wet season by approximately 1 dB. One might attribute this difference to the microphysical processes involved, such as that during the wet (dry) season, the melting layer is mainly driven by warm rain (ice microphysics) processes (Dolan et al., 2013).

One of the main differences in the cloud structure between the wet and dry season relies on the cloud top altitudes. Indeed, during the dry season, clouds can easily reach 16-17 km in the tropics compared to only 13-14 km during the wet season. Therefore, the microphysical processes for negative temperatures are distributed over two different thickness layers and moisture profiles. In this cloud region, both aggregates and ice crystals mainly grow by vapor diffusion (Houze, 1997). Although they present quite

similar distributions, they both spread at about a 1.5 km interval difference in altitude. Additionally, the $Z_{DR}$ values associated with aggregates and ice crystals are generally slightly higher than those retrieved



in DR09 or GR15. However, this result is consistent with the study of Wendisch et al (2016) that identified shaped plates of aggregates/crystals in the anvil outflow with in situ airplane observations.

585        **2) Convective region**

Figure 14 presents a comparison of pairs of convective microphysical species between the wet and dry seasons. As aforementioned in section 5, the dry season is composed of 7 hydrometeor types compared to 6 for the wet season. While the rainy season only has a graupel microphysical species, the dry season allows distinguishing between low- and high-density graupel. Therefore, the graupel microphysical

species defined during the wet season has been associated with the low-density graupel of the dry season to make this comparison possible.

Convective regions are characterized by three different rainfall regimes: light, moderate and heavy rain. Overall, the $Z_H$, $Z_{DR}$, and $K_{DP}$ distributions associated with the dry season are generally shifted towards higher values. The dry season is known to exhibit the most intense convective cells (Machado et al,

2018). Their corresponding precipitation formation mechanism is generally dominated by ice microphysical processes, wherein the melting of graupel particles lead to large raindrops (Rosenfeld and Ulbrich, 2003; Dolan et al., 2013). One can see here that although growth by coalescence could be very efficient during the wet season, the ice microphysical processes outweigh the production of larger raindrops.

Overall, the combination of the wet season graupel microphysical species with the dry season low-density graupel makes sense in Figure 14. Indeed, they have almost the same polarimetric range distributions and are in agreement with each other. By contrast, the high-density graupel signatures are



correlated with high $Z_H$, $Z_{DR}$, and $K_{DP}$ values and low $\rho_{HV}$ values. As mentioned in section 5.b.2, high-density graupel would have been associated with a mixture of wet graupel/small hail. Nevertheless,

these three related graupel categories are even consistent with the DR09 T-matrix definitions.

The main discrepancy between the aggregate and ice crystal microphysical species concerns their altitude definitions, wherein the dry season allows generating these hydrometeor types at higher altitudes. Systematically, the aggregate and ice crystal $Z_H$ and $Z_{DR}$ distributions are shifted to higher values during the wet season. These shifts may be due to an unreliable estimation of the attenuation

correction or explained by the results of Rosenfeld et al (1998) and Giangrande et al (2016). Both of these studies showed that during the dry season, updrafts are more intense and, therefore, do not allow enough time for small ice crystals to properly develop. Additionally, Williams et al (2002) or even Cecchini et al (2016) highlighted the impact of aerosol concentrations on the microphysical development of cloud particles. According to these studies, during the dry season, the higher the aerosol

concentration is, the more the coalescence process is suppressed (thus, leading to smaller particles). In terms of aerosol concentrations, the wet Amazonian season is known to be much cleaner than the dry season (Artaxo et al. 2002).

## 7. Conclusions

Based on an innovative clustering approach, the first hydrometeor classification for Amazon tropical-equatorial precipitation systems has been realized by using research X-band DPOL radar deployed during both the GoAmazon2014/5 and ACRIDICON-CHUVA field experiments. The AHC method was broadly equivalent to GR15 and built using $Z_H$, $Z_{DR}$, $K_{DP}$ and $p_{HV}$ polarimetric radar variables together





with temperature information extracted from sounding balloons. The clustering approach allowed

gathering of polarimetric radar observations that exhibit similarities amongst themselves within both

wet and dry seasons and both stratiform and convective regions. Sensitivity analysis during the wet

season was performed through different linkage rules and showed that both the weighted and Ward

linkage rules were the most suitable for this hydrometeor classification task. In this regard, a novel

approach was tested to improve the 0 °C hydrometeor layer representation within the convective region.

While the 0 °C isotherm region is generally binarily represented, one can allow the liquid water content

to overpass this region by setting simple rules. The final representation showed a realistic distribution

and created new perspectives to respect polarimetric radar signatures as much as possible.

The AHC clustering outputs for both the wet and dry seasons and the stratiform and convective regions

were investigated over the Manaus region with the complete datasets collected during 2014. Although

previous studies were conducted for different latitudes and/or wavelengths, the retrieved hydrometeor

types were found to be generally in agreement. Overall, typical cloud microphysical distributions within

the stratiform precipitation regimes are characterized by five hydrometeors: drizzle/light rain, rain, wet

snow, aggregates, and ice crystals. On the other hand, convective regions exhibit more diversified

microphysical populations with six (seven) retrieved hydrometeor types for the wet (dry) season: light

rain, moderate rain, heavy rain, low-density graupel, (high-density graupel), aggregates, and ice

crystals.

The present study also highlighted the potential of the clustering approach in comparison to a more

"classical" supervised fuzzy logic algorithm. For instance, the clustering results showed a better ability

to delimit and distinguish the bright band region. The AHC method also allowed exploiting the higher

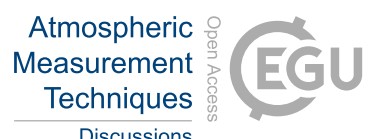

sensitivity of the X-band radar and permitted retrieving three different rainfall regimes by exhibiting

light, moderate, and heavy intensities.

The retrieved labelled clusters allowed making comparisons of the dominant microphysical species

involved during both the wet and dry seasons of Brazilian tropical precipitation systems. Thus, the main

discrepancy relies on the presence of one more microphysical species within the convective region of

the dry season, defined as high-density graupel. This microphysical species is probably the result of a

deeper convection associated with precipitation systems that occur during this period of the year.

Overall, the dry season $Z_H$, $Z_{DR}$, and $K_{DP}$ distribution shapes were quite similar to those of the rainy

period; however, the distributions were shifted towards higher (lower) values for positive (negative)

temperatures. The different rainfall intensities associated with the dry season generally exhibited higher

$Z_H$, $Z_{DR}$, and $K_{DP}$ values than those during the wet season, leading us to believe that ice microphysical

processes outweigh warm rain microphysical mechanisms. Finally, the retrieved tropical microphysical

species distribution showed that both aggregates and ice crystals were shifted towards higher radar

observable values in comparison to the mid-latitude X-band definition. These signatures might be due to

the presence of a higher humidity amount within tropical regions, which may allow more dendritic-plate

growth of aggregates and ice crystals microphysical species.

Although the year 2014 was representative and complied with typical tropical precipitation events, the

present study could be strengthened by an extended dataset as well as the use of in situ observations for

validation tasks. Nevertheless, this first detailed analysis of dominant hydrometeor distributions within

tropical precipitation systems is promising and could also be extended to other radar frequencies and



operational DPOL radars. Such improvements could be useful to identify key microphysical parameters for nowcasting issues, which are expected to be investigated in the near future through both the SOS-CHUVA (Brazil) and RELAMPAGO (Argentina) research projects. In this regard, the clustering methodology could be enhanced by taking into account the Doppler velocities to explore the
microphysical processes involved within vigorous updraft/downdraft regions of the cloud. Finally, these results could also be helpful in evaluating the microphysical parameterization schemes used within high-resolution numerical weather prediction models.


**Acknowledgements**

The authors would like to especially thank Jacopo Grazioli for fruitful discussions about the clustering method that helped refine the ideas developed in this study. The contribution of the first author was supported by the São Paulo Research Foundation (FAPESP) under grants 2016/16932-8 and 2015/14497-0 for the SOS-CHUVA project.


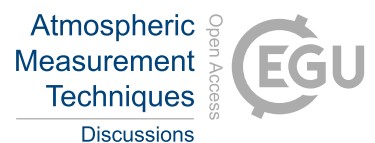

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





## List of Tables

*Table 1: X-band dual-polarization radar characteristics*

*Table 2: Distance formulas for the weighted, centroid and Ward linkage rules.*

*Table 3: Confusion matrix comparing the clustering outputs from the stratiform region of the wet season and hydrometeor species retrieved from the adapted fuzzy logic.*

*Table 4: Same as Table 3, but for the convective region of the wet season.*

*Table 5: Same as Table 3, but for the stratiform region of the dry season.*

*Table 6: Same as Table 3, but for the stratiform region of the dry season.*

## List of figures

*Figure 1: Schematic representation of the different hydrometeor classification techniques and their principal associated benchmarks.*

*Figure 2: (a) Geographical localization of the GoAmazon2014/5 and ACRIDICON-CHUVA experiments. (b) X-band DPOL radar coverage and its associated topography.*

*Figure 3: Evolution of the variance explained for different clustering linkage rules. Each linkage method is subdivided in terms of stratiform (dashed line) and convective (solid line) regions. The orange vertical span highlights the interval potentially associated with the optimal number of clusters.*

*Figure 4: X-band DPOL radar observables and the corresponding retrieved hydrometeor classification outputs at 12:07 UTC on 21 February 2014, along the azimuth 290°. DPOL radar observables are shown in panels: (a) $Z_H$, (b) $Z_{DR}$, (c) $K_{DP}$, and (d) $\rho_{HV}$. Comparisons of retrieved hydrometeors for clustering outputs based on (e) weighted, (f) centroid, and (g) Ward linkage rules and (h) fuzzy logic scheme outputs. In panels (e)-(f)-(g), each number corresponds to a different cluster. 'S' stands for stratiform regimes, whereas 'C' is for convective regimes.*

*Figure 5: Same as Figure 4, but for 13:57 UTC on 13 February 2014, along the azimuth 200°.*





**Figure 6:** *Clustering hydrometeor classification retrieved from the X-band radar at 12:07 UTC on 21 February 2014, along the azimuth 290°. (a) With temperature constraint, (b) without temperature constraint.*

**Figure 7:** *Violin plot of cluster outputs retrieved for the stratiform regime of the wet season (DZ: drizzle, RN: rain, WS: wet snow, AG: aggregates, IC: ice crystals). The thick black bar in the centre represents the interquartile range, and the thin black line extended from it represents the 95 % confidence intervals, while the white dot is the median.*

**Figure 8:** *Same as Figure 7, but for the convective regime of the wet season (LR: light rain, MR: moderate rain, HR: heavy rain, GR: graupel, AG: aggregates, IC: ice crystals).*

**Figure 9:** *X-band DPOL radar observables and the corresponding retrieved hydrometeor classification outputs at 21:26 UTC on 08 September 2014, along the azimuth 200°. DPOL radar observables are shown in panels: (a) $Z_H$, (b) $Z_{DR}$, (c) $K_{DP}$, and (d) $\rho_{HV}$. Comparisons of retrieved hydrometeors for clustering outputs based on (e) weighted linkage rules and (f) the fuzzy logic scheme. In panels (e)-(f), each number corresponds to a different cluster. 'S' stands for the stratiform region, whereas 'C' is for the convective region.*

**Figure 10:** *Same as Figure 7, but for the stratiform regime of the dry season (DZ: drizzle, RN: rain, WS: wet snow, AG: aggregates, IC: ice crystals).*

**Figure 11:** *Same as Figure 9, but for an RHI at 18:16 UTC on 06 October 2014, along the azimuth 200°.*

**Figure 12:** *Same as Figure 7, but for the convective regime of the dry season (LR: light rain, MR: moderate rain, HR: heavy rain, LDG: low-density graupel, HDG: high-density graupel, AG: aggregates, IC: ice crystals).*

**Figure 13:** *Violin plot comparison of pairs of stratiform hydrometeor types between the wet and dry seasons (DZ: drizzle, RN: rain, WS: wet snow, AG: aggregates, and IC: ice crystals).*

**Figure 14:** Same as Figure 13, but for the convective precipitation regime (LR: light rain, MR: moderate rain, HR: heavy rain, LDG: low-density graupel, HDG: high-density graupel, AG: aggregates, and IC: ice crystals).

10.5194/amt-2018-174
Atmospheric Measurement Techniques
2018-09-19



| Location | (3.21ºS; 60.6ºW; 60.9m) |
|---|---|
| Radar type | Pulsed |
| Polarization | H-V orthogonal |
| Transmission/reception | Simultaneous |
| Antenna | 1.8 m diameter, 1.3º 3 dB beamwidth |
| Antenna gain | 43 dB |
| Frequency | 9.345 GHz |
| Maximum range detection | 100 km |
| Range resolution | 200 m |
| 10 min PPI elevation angles | 0.5°/1.3º/2.1º/3.2º/4.3º/5.6º/7.1º/8.8º/10.8º/13.0º/15.6º/18.5º/21.8º/25.6º/30.0º |

*Table 1: X-band dual-polarization radar characteristics*

| Linkage method | Distance formula for $d(S \cup T, V)$ |
|---|---|
| Weighted | $\frac{d(S,V)+d(T,V)}{2}$ |
| Centroid | $\sqrt{\frac{n_S d(S,V)+n_T d(T,V)}{n_S+n_T} - \frac{n_S n_T d(S,T)}{(n_S+n_T)^2}}$ |
| Ward | $\sqrt{\frac{(n_S+n_V)d(S,V)+(n_T+n_V)d(T,V)-n_V d(S,T)}{n_S+n_T+n_V}}$ |



*Table 2: Distance formulas for the weighted, centroid and Ward linkage rules.*






| TYPE | DZ | RN | MH | WS | AG | LDG | HDG | VI | CR |
|---|---|---|---|---|---|---|---|---|---|
| **1S** | 38.64 % | 0.01 % | 0.00 % | 10.34 % | 32.91 % | 1.31 % | 0.00 % | 4.47 % | 12.34 % |
| **2S** | 0.02 % | 0.21 % | 0.00 % | 43.51 % | 42.66 % | 11.91 % | 0.00 % | 0.02 % | 1.67 % |
| **3S** | 64.36 % | 27.55 % | 0.21 % | 7.88 % | 0.00 % | 0.00 % | 0.00 % | 0.00 % | 0.00 % |
| **4S** | 5.75 % | 7.27 % | 0.02 % | 86.02 % | 0.53 % | 0.11 % | 0.00 % | 0.03 % | 0.27 % |
| **5S** | 98.04 % | 0.00 % | 0.27 % | 1.68 % | 0.00 % | 0.00 % | 0.00 % | 0.00 % | 0.00 % |

***Table 3:*** *Confusion matrix comparing the clustering outputs from the stratiform region of the wet season and hydrometeor species retrieved from the adapted fuzzy logic.*



| TYPE | DZ | RN | MH | WS | AG | LDG | HDG | VI | CR |
|---|---|---|---|---|---|---|---|---|---|
| **6C** | 77.00 % | 21.70 % | 0.99 % | 0.31 % | 0.00 % | 0.00 % | 0.00 % | 0.00 % | 0.00 % |
| **7C** | 0.00 % | 0.16 % | 0.00 % | 21.69 % | 7.70 % | 69.01 % | 1.44 % | 0.00 % | 0.00 % |
| **8C** | 0.78 % | 2.70 % | 0.02 % | 27.24 % | 44.51 % | 23.71 % | 0.00 % | 0.27 % | 0.77 % |
| **9C** | 0.10 % | 0.00 % | 0.00 % | 9.86 % | 55.90 % | 5.83 % | 0.00 % | 9.15 % | 19.16 % |
| **10C** | 96.47 % | 0.14 % | 1.46 % | 1.92 % | 0.00 % | 0.00 % | 0.00 % | 0.00 % | 0.00 % |
| **11C** | 31.42 % | 62.98 % | 1.24 % | 4.36 % | 0.00 % | 0.00 % | 0.00 % | 0.00 % | 0.00 % |

***Table 4:*** *Same as Table 3, but for the convective region of the wet season.*






| TYPE | DZ | RN | MH | WS | AG | LDG | HDG | VI | CR |
|---|---|---|---|---|---|---|---|---|---|
| 1S | 76.30 % | 22.17 % | 0.10 % | 1.43 % | 0.00 % | 0.00 % | 0.00 % | 0.00 % | 0.00 % |
| 2S | 92.32 % | 4.36 % | 0.65 % | 2.63 % | 0.02 % | 0.00 % | 0.00 % | 0.01 % | 0.00 % |
| 3S | 0.25 % | 0.00 % | 0.00 % | 2.65 % | 41.61 % | 2.19 % | 0.00 % | 21.18 % | 32.12 % |
| 4S | 0.97 % | 1.30 % | 0.00 % | 49.30 % | 18.46 % | 26.83 % | 0.23 % | 0.44 % | 2.48 % |
| 5S | 0.30 % | 0.03 % | 0.00 % | 8.28 % | 68.48 % | 3.99 % | 0.00 % | 5.29 % | 13.62 % |

**Table 5:** *Same as Table 3, but for the stratiform region of the dry season.*


| TYPE | DZ | RN | MH | WS | AG | LDG | HDG | VI | CR |
|---|---|---|---|---|---|---|---|---|---|
| 6C | 73.71 % | 23.34 % | 2.60 % | 0.34 % | 0.00 % | 0.00 % | 0.00 % | 0.00 % | 0.00 % |
| 7C | 21.61 % | 73.56 % | 1.00 % | 3.83 % | 0.01 % | 0.00 % | 0.00 % | 0.00 % | 0.00 % |
| 8C | 0.07 % | 0.01 % | 0.00 % | 5.62 % | 51.01 % | 2.70 % | 0.00 % | 12.72 % | 27.87 % |
| 9C | 0.16 % | 2.32 % | 0.00 % | 27.80 % | 7.41 % | 60.40 % | 1.86 % | 0.00 % | 0.04 % |
| 10C | 0.79 % | 0.17 % | 0.00 % | 13.48 % | 51.19 % | 30.91 % | 0.00 % | 0.83 % | 2.63 % |
| 11C | 0.00 % | 15.29 % | 0.51 % | 64.19 % | 0.19 % | 11.4 % | 7.72 % | 0.00 % | 0.00 % |
| 12C | 97.19 % | 0.00 % | 0.41 % | 2.34 % | 0.06 % | 0.00 % | 0.00 % | 0.01 % | 0.00 % |

**Table 6:** *Same as Table 3, but for the convective region of the dry season.*




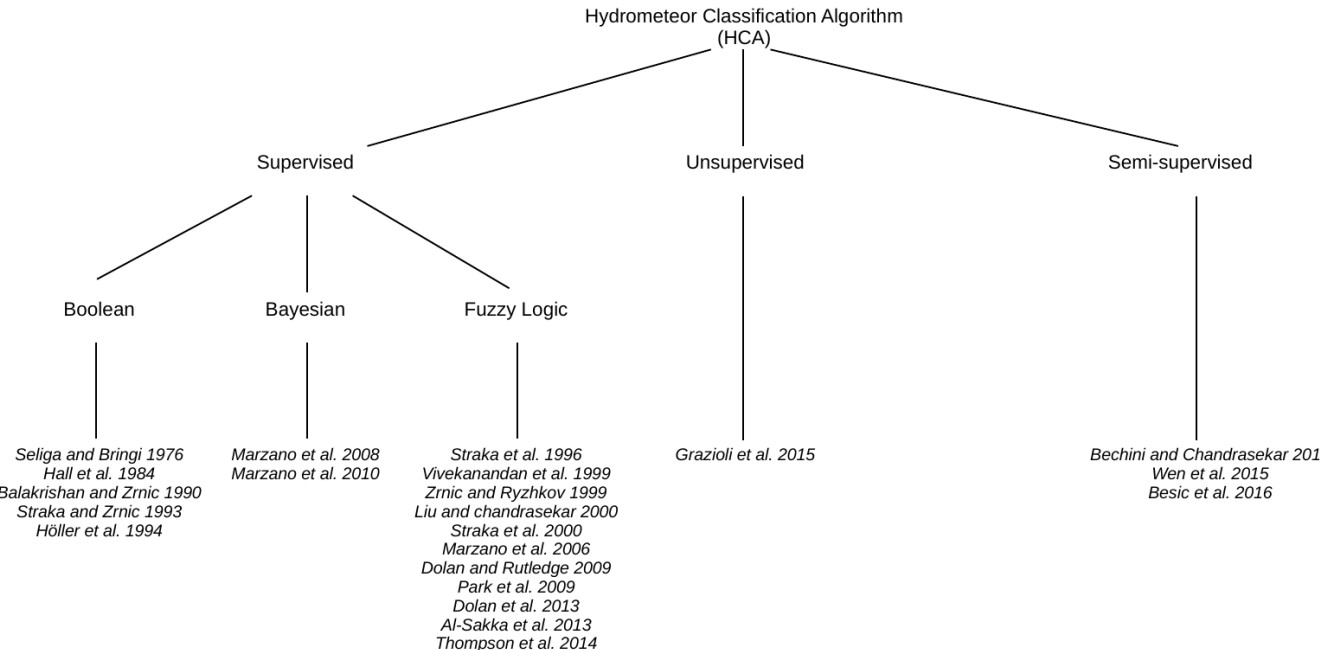

***Figure 1:*** *Schematic representation of the different hydrometeor classification techniques and their*
*principal associated benchmarks.*






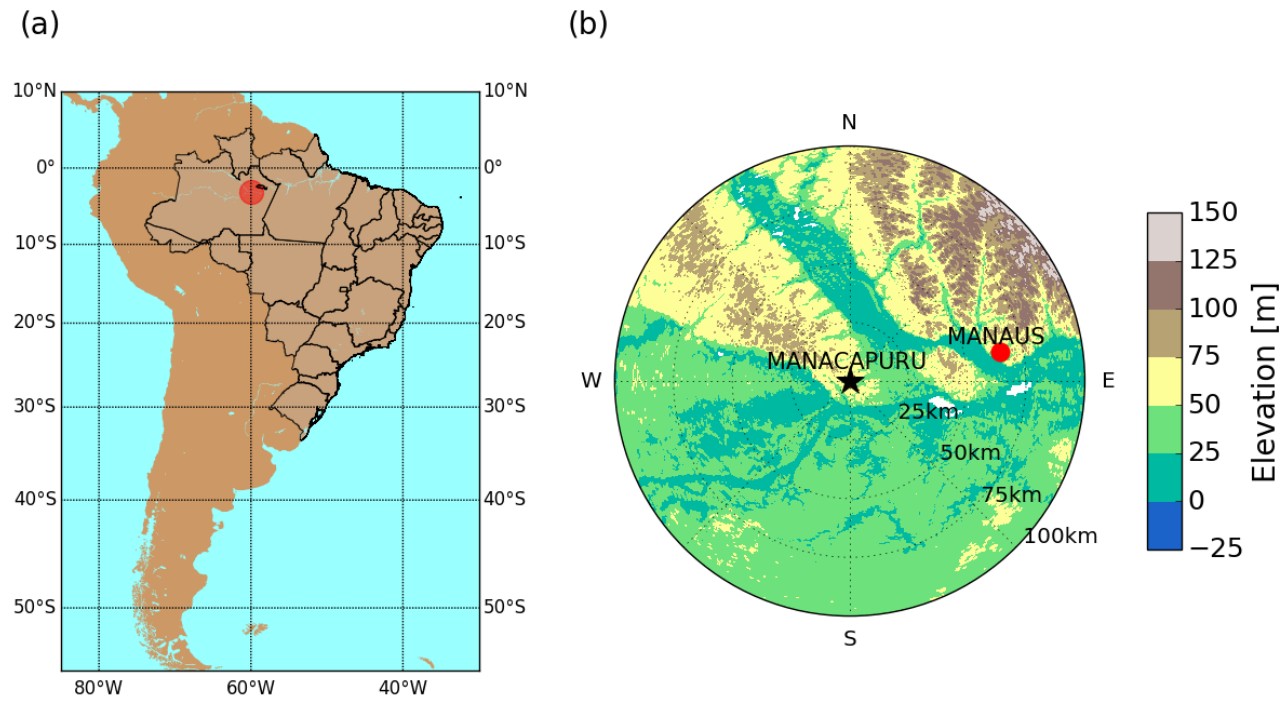

***Figure 2:*** *(a) Geographical localization of the GoAmazon2014/5 and ACRIDICON-CHUVA experiments. (b) X-band DPOL radar coverage and its associated topography.*











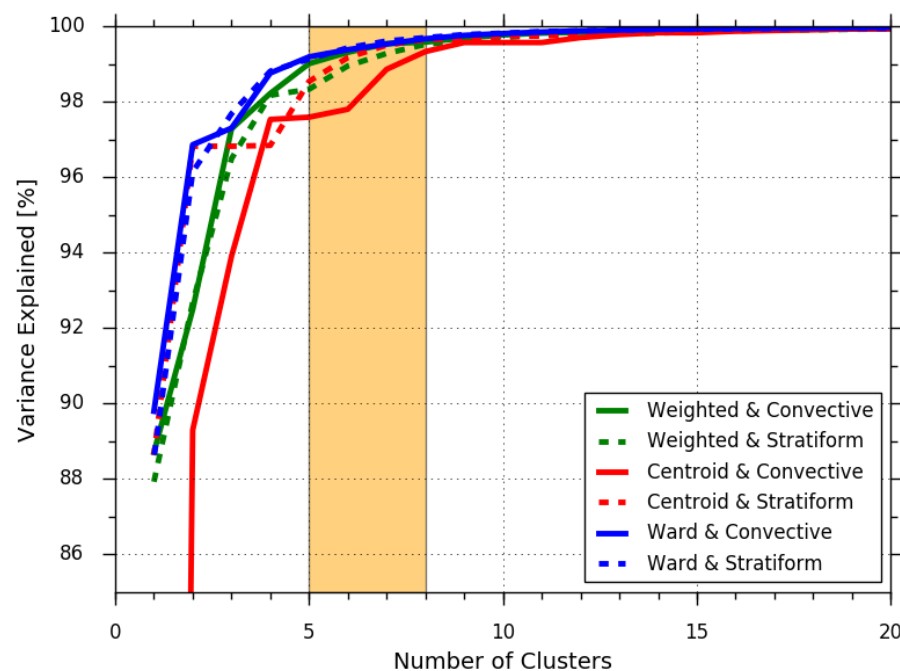


***Figure 3:*** *Evolution of the variance explained for different clustering linkage methods. Each linkage method is subdivided in terms of stratiform (dashed line) and convective (solid line) regions. The orange vertical span highlights the interval potentially associated with the optimal number of clusters.*





***Figure 4:*** *X-band DPOL radar observables and corresponding retrieved hydrometeor classification outputs at 12:07 UTC on 21 February 2014, along the azimuth 290°. DPOL radar observables are shown in panels (a) $Z_H$, (b) $Z_{DR}$, (c) $K_{DP}$, and (d) $\rho_{HV}$. Comparisons of retrieved hydrometeors for clustering outputs based on (e) weighted, (f) centroid, and (g) Ward linkage rules and (h) fuzzy logic scheme outputs. In panels (e)-(f)-(g), each number corresponds to a different cluster. 'S' stands for stratiform regimes, whereas 'C' is for convective regimes.*



**Figure 5:** *Same as Figure 4, but for 13:57 UTC on 13 February 2014, along the azimuth 200°.*




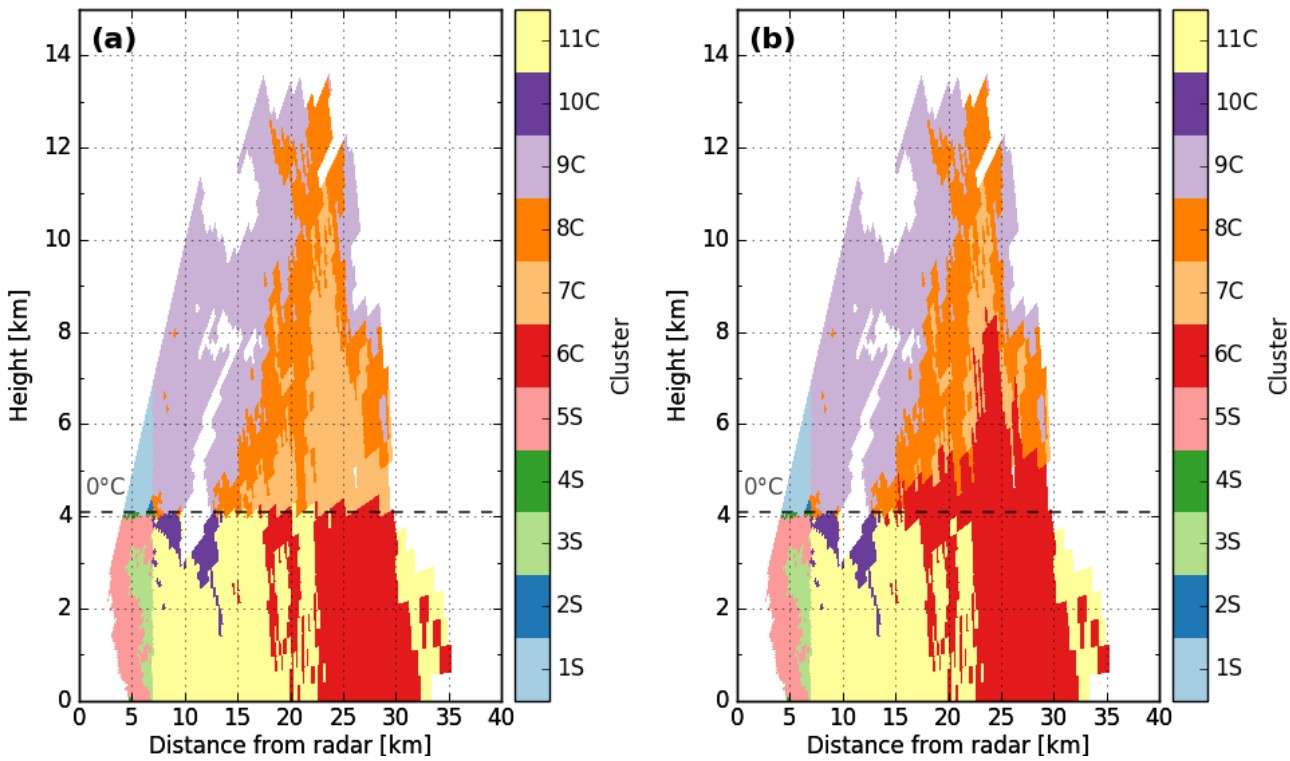

***Figure 6:*** *Clustering hydrometeor classification retrieved from the X-band radar at 12:07 UTC on 21*
*February 2014, along the azimuth 290°. (a) With temperature constraint, (b) without temperature*
*constraint.*







**Figure 7:** *Violin plot of cluster outputs retrieved for the stratiform regime of the wet season (DZ: drizzle, RN: rain, WS: wet snow, AG: aggregates, IC: ice crystals). The thick black bar in the centre represents the interquartile range, and the thin black line extended from it represents the 95 % confidence intervals, while the white dot is the median.*





**Figure 8:** *Same as Figure 7, but for the convective regime of the wet season (LR: light rain, MR: moderate rain, HR: heavy rain, GR: graupel, AG: aggregates, IC: ice crystals).*





**Figure 9:** *X-band DPOL radar observables and corresponding retrieved hydrometeor classification outputs at 21:26 UTC on 08 September 2014, along the azimuth 200°. DPOL radar observables are shown in panels (a) $Z_H$, (b) $Z_{DR}$, (c) $K_{DP}$, and (d) $\rho_{HV}$. Comparisons of the retrieved hydrometeor for clustering outputs based on (e) weighted linkage rules and (f) the fuzzy logic scheme. In panels (e)-(f), each number corresponds to a different cluster. 'S' stands for the stratiform region, whereas 'C' is for the convective region.*





**Figure 10:** *Same as Figure 7, but for the stratiform regime of the dry season (DZ: drizzle, RN: rain, WS: wet snow, AG: aggregates, IC: ice crystals).*



**Figure 11:** *Same as Figure 9, but for an RHI at 18:16 UTC on 06 October 2014, along the azimuth* 1240 *200°.*





**Figure 12:** *Same as Figure 7, but for the convective regime of the dry season (LR: light rain, MR: moderate rain, HR: heavy rain, LDG: low-density graupel, HDG: high-density graupel, AG: aggregates, IC: ice crystals).*







**Figure 13:** Violin plot comparison of pairs of stratiform hydrometeor types between the wet and dry seasons (DZ: drizzle, RN: rain, WS: wet snow, AG: aggregates, and IC: ice crystals).



**Figure 14:** Same as Figure 13, but for the convective precipitation regime (LR: light rain, MR: moderate rain, HR: heavy rain, LDG: low-density graupel, HDG: high-density graupel, AG: aggregates, and IC: ice crystals).



**APPENDIX A:** Wet and Dry Season cluster centroids

| Cluster | Label | $Z_H$ [dBZ] | $Z_{DR}$ [dB] | $K_{DP}$ [degree/km] | $P_{HV}$ [-] | $\Delta z$ [km] |
|---------|-------|-------------|---------------|----------------------|--------------|-----------------|
| 1S | Ice Crystals Small Aggregates | 17.18 | 1.17 | 0.21 | 0.98 | + 2.23 |
| 2S | Aggregates | 27.09 | 1.31 | 0.27 | 0.97 | + 1.25 |
| 3S | Rain | 27.28 | 1.43 | 0.10 | 0.97 | - 2.49 |
| 4S | Wet Snow | 27.54 | 1.83 | 0.07 | 0.95 | - 0.10 |
| 5S | Drizzle | 13.84 | 1.21 | 0.02 | 0.99 | - 3.00 |
| 6C | Heavy Rain | 44.18 | 2.09 | 1.88 | 0.98 | - 2.81 |
| 7C | Graupel | 36.28 | 0.74 | 0.34 | 0.98 | + 2.76 |
| 8C | Aggregates | 28.94 | 0.75 | 0.20 | 0.98 | + 2.32 |
| 9C | Ice Crystals Small Aggregates | 17.62 | 0.91 | 0.22 | 0.97 | + 3.07 |
| 10C | Light Rain | 13.21 | 0.68 | 0.14 | 0.96 | - 2.81 |
| 11C | Moderate Rain | 31.09 | 1.39 | 0.50 | 0.98 | - 2.74 |

***Table A.1:*** *Cluster centroids for the wet season.*





| Cluster | Label | $Z_H$ [dBZ] | $Z_{DR}$ [dB] | $K_{DP}$ [degree/km] | $P_{HV}$ [-] | $\Delta z$ [km] |
|---|---|---|---|---|---|---|
| 1S | Rain | 31.43 | 1.27 | 0.25 | 0.98 | - 3.12 |
| 2S | Drizzle | 20.66 | 0.89 | 0.07 | 0.98 | - 3.16 |
| 3S | Ice Crystals Small Aggregates | 13.61 | 0.11 | 0.06 | 0.98 | + 3.65 |
| 4S | Wet Snow | 29.18 | 0.85 | 0.17 | 0.93 | + 1.40 |
| 5S | Aggregates | 19.65 | 0.71 | 0.11 | 0.98 | + 3.04 |
| 6C | Heavy Rain | 46.7 | 2.38 | 3.12 | 0.97 | - 2.90 |
| 7C | Moderate Rain | 34.18 | 1.24 | 1.06 | 0.97 | - 2.82 |
| 8C | Ice Crystals Small Aggregates | 16.69 | 0.43 | 0.11 | 0.97 | + 3.85 |
| 9C | Low-Density Graupel | 36.79 | 0.78 | 0.59 | 0.97 | + 1.96 |
| 10C | Aggregates | 24.75 | 0.45 | 0.18 | 0.98 | + 3.20 |
| 11C | High-Density Graupel | 46.36 | 2.20 | 2.50 | 0.94 | + 0.50 |
| 12C | Light Rain | 14.47 | 0.27 | 0.21 | 0.97 | - 2.89 |

***Table A.2:*** *Cluster centroids for the dry season.*
