# Peer review of "X-band dual-polarization radar-based hydrometeor classification for Brazilian tropical precipitation systems"

_Atmospheric Measurement Techniques, 2018_

## Referee Comment (RC2) · Anonymous Referee #2 · 30 Nov 2018

This is my second review of this manuscript. Overall, the authors have addressed the previous comments to the reviewer's satisfaction. However, looks like some of the responses are not really well integrated in the main manuscript. The authors should revise the manuscript thoroughly according to their own responses.

For example, the authors have rephrased the description of using the melting layer as a parameter to detect liquid-ice delineation (lines 146-147 in the revised submission). But the content after this has not been revised accordingly (lines 147-151), which is still misleading.

The response about calibration of ZDR should be included in the main text as well.

[Figure]

In addition, the authors stated that the attenuation correction would not significantly change the results, which is very subjective. The authors referred to Dolan and Rutledge (2009) regarding this point. But Dolan and Rutledge (2009) was devised from a very different aspect (i.e., scattering simulation and fuzzy logic application, no clustering).

Extra and careful attention should be paid to all these points. Other than these, I think this manuscript is ready for publication.

---

## Author Comment (AC1) · 22 Dec 2018

The comment was uploaded in the form of a supplement:
https://www.atmos-meas-tech-discuss.net/amt-2018-174/amt-2018-174-AC1-supplement.pdf

---

## Author Comment (AC2) · 22 Dec 2018

**Responses to Reviewer #1**

We thank you for the thoughtful comments and changes suggested in your review of our manuscript. Our point-to-point responses are developed hereafter, along with an indication of changes made in the revised version of the text.

**1 - Despite the authors' response to my question last time, I still believe that separating convective and stratiform echoes, and therefore hydrometeor types, is unnecessary. Have the authors compared (via the violin plots; I really appreciate this presentation method in the paper!) stratiform and convective aggregates, ice crystals, rain, etc.? I also wonder if these differences would pop out of the algorithm at all if they were all included together.**

First, we would like to mention that the present study have been initially developed without making any separation between stratiform/convective echoes. In this respect, results have been presented at the 38th Conference on Radar Meteorology held in Chicago in 2018 (Ribaud et al, 2018 - https://ams.confex.com/ams/38RADAR/webprogram/38RADAR.html), where it is possible to get access to the "merged" Ice Crystals, Aggregates,…DPOL characteristics for each radar observable.

Nevertheless, we must admit that both the GoAmazon2014/5 and the ACRIDICON-CHUVA field experiments that took place in the Amazon basin during the wet and dry seasons, offer the possibility to explore new microphysical aspects (such as DPOL differences), which is an important objective of those two research programs. In addition, we should specify that except the study of Dolan et al (2013) that explore such DPOL seasonal differences in Australia, nothing have been done in another region (in terms of HCA), offering here an unique opportunity to investigate those microphysical aspects and inter-compare them.

To achieve this goal a stratiform-convective differentiation is needed, due to thermodynamic differences that characterized those regions, but more specifically in the Amazon basin because of aerosol loading that (in-)directly impacts onto different microphysical aspects as related, for instance, by Machado et al (2018). With this regard, and to our point of view, it was more appropriate to methodologically run the clustering routine by separating those different echoes so as to investigate as close as possible their microphysical characteristics.

In the end, by inter-comparing resulting stratiform and convective hydrometeor types and especially those coexisting within both cloud regions, we agree with the reviewer that probably Ice Crystals or even Aggregates could be possibly merged. Making such combination would nevertheless require significant amount of work in order to reanalyse the results and modify accordingly the structure and diagrams of the paper. Also, we believe that these modifications would be difficult to make within the time frame allowed to revise the paper but, more importantly, we already know that the outcomes of such combination would be extremely limited.

With this regard, discussions about microphysical coexistence between the two cloud types have been included in the section 6-a of the revised manuscript:

"*Note that the present clustering method has been distinctly subdivided into stratiform and convective regions. Although they are characterized by different thermodynamic structures (Houze, 1997), the stratiform and convective regions may be related in terms of microphysical distributions, such as ice particles which might be ejected from the top of an active convective cell into the upper part of the stratiform region. This microphysical continuity could be further considered either by merging stratiform and convective hydrometeor types that present close*

*DPOL characteristics (Figures 7-8-10-12), or by implementing an a posteriori continuity analysis."*

We are confident that you will understand our choice and hope that you will allow this paper to be published despite the fact that few hydrometeor types are not merged between stratiform and convective echoes.

**2 - I'm also not completely sold on the explanation for differences between the wet and dry season in terms of their radar characteristics of different hydrometeor types and physical underpinnings. Obviously knowing microphysically exactly what leads to the differences is difficult to verify and perhaps even beyond the scope of this paper. This is definitely very interesting to think about. Is there a possibility of eventually bringing in aerosol / CCN information from the field projects in the area?**

We agree. However the scope of the present paper (AMT) was to introduce the methodology and the first Brazilian hydrometeor classification without any defined threshold (unsupervised). In this respect, the classification has been applied to characterize the difference between wet and dry seasons and for stratiform and convective clouds. Only these features made the paper already very long. That is why we did not explore further those microphysical differences.

From both the GoAmazon2014/5 and ACRIDICON-CHUVA projects, it could be however interesting to bring aerosol/CCN information. Also, we would like to mention that significant advances have been realized in terms of aerosol-cloud interactions over the Amazonas region, opening new perspectives in tropical clouds understanding. Therefore the present classification will be applied in near future to study rainfall-aerosol interaction, as well as the application for convective clouds and their respective evolutions.

While this is outside the scope of the present paper, this possibility has been mentioned in the concluding section of the revised manuscript.

*"Although the year 2014 was representative and complied with typical tropical precipitation events, the present study could be strengthened by an extended dataset as well as the use of i) in situ observations for validation tasks and ii) aerosols information to investigate microphysical differences between the wet and dry season."*

**3 - Pg 6, ln 125: I think "dangerousness" could be sufficiently replaced by "danger".**
Corrected as suggested.

**4 - Pg 7, ln 128: I'm not sure I understand what "climatically radar-dependent" means.**
Please understand "regionally radar-dependent" here.

**5 - Pg 8, ln 166 / Fig. 2: Is there any significant blockage to be concerned with?**
Not at all. The area scanned by the radar is rather flat ($\Delta h \sim 100m$) and nothing has been noticed through the dataset.

**6 - Pg 9, ln 173: Was there reflectivity calibration performed during these two different seasons to look for any potential drift that might account for differences between the seasons?**
The radar calibration has been done carefully and in the same manner during both the wet and dry seasons.

**7 - Pg10, ln 203: Has the acronym AHC been introduced yet?**
Yes, it has already been defined in the abstract section.

**8 - Pg 11, ln 222: I don't follow what is meant by "The first four components of each object are based on the minimum-maximum boundaries rule." What are the objects?**
The objects refer to radar observations randomly chosen through the whole precipitation events database and where each observation is defined as $x = \{Z_H, Z_{DR}, KDP, \rho_{HV}, \Delta z\}$.

**9 - Pg 13, ln 263: It occurs to me to ask what the actual sensitivity to the linkage rule is, and if it is the only "tuning" parameter in the classification algorithm?**
You are right. The linkage rule is not the only "tuning" parameter. For instance, one can consider to give more/less weight to the temperature information ($\Delta z$) by distributing its values into a different range space. Grazioli et al 2015 already tested this specific point.

**10 - Pg 13, ln 269: Is an "observation" a single radar gate?**
Yes, it is.

**11 - Pg 13, ln 270: Is the temperature information constant across the domain?**
Yes, we considered and applied a constant temperature over the domain.

**12 - Pg 15, ln 309: Do you perhaps mean "below" instead of "on top" of the aggregate mixture?**
Corrected as suggested.

**13 - Pg 18, top: What happens if no temperature is included in the clustering?**
We would have liked to directly exclude the temperature information from the initial clustering steps. However, clustering outputs appeared to do not make sense anymore. For instance ice crystals were merged with light rain, or only one "liquid" cluster was defined in the convective core with altitudes spanning from ~ 0 to 9 km... That is why we need to deal with temperature information first.

**14 - Pg 23, ln 475: Is there a word/phrase missing here (such as "the same as")? Otherwise this is confusing that the AHC is putting out four clusters as fuzzy logic outputs.**
Corrected as suggested.

**15 - Pg 24, ln 480: Not to be picky, but Table 6 shows the value is 2.6% which exceeds 2%, so I would round up and say "none of the clustering outputs exceed 3%".**
Corrected as suggested.

**16 - Pg 26, ln 536: Change to "rain was four times" (e.g. add 's').**
Corrected.

**17 - Pg 27, ln 551: It is a little odd to me to be talking about "drizzle" and breakup processes. Perhaps using the "drizzle/light rain" nomenclature above in line 549 would be helpful.**
Please now reads:
*"These polarimetric signatures might be attributed to the evaporation and collisional processes that tend to reduce the particle diameters (Kumjian and Ryzhkov 2010; Penide et al, 2013)."*

**18 - Pg 27, ln 553: "…presents a more favourable environment…" For what?**
You are right. It was not clear and the sentence has been removed.

**19 - Pg 27, ln 558: "The general wet rain microphysical species distribution thus still contains drizzle/light rain observations. This puzzling rain partitioning…" I am not sure what is being conveyed here.**
Please now reads:
*"The general wet rain microphysical species distribution thus still contains drizzle/light rain observations, which might be due to the different cloud cover patterns associated with stratiform echoes during the two seasons."*

**20 - Pg 28, "the melting layer is mainly driven by warm rain". This does not make sense; what does warm rain have to do with the melting layer? I would think the characteristics of the melting layer, and resulting radar bright band which is what we are really talking about here, is driven by the ice particles aloft which are melting. Therefore, in this context, larger Zdr values might be related to larger particles in the ice phase aloft melting?**
You are totally right. The discussion has been modified and please now reads:
*"According to the study of Wang et al. (2018) which put emphasis onto mature Mesoscale Convective System events during the GoAmazon2014/5 experiment, the wet season always presents stronger bright band signatures that might be attributed to more prominent aggregation processes. Indeed, the moist conditions in midlevels could promote more ice growth in the stratiform regions (as compared to the dry season) and could lead to stronger bright band signatures when those aggregates melt."*

Wang, D., Giangrande, S. E., Bartholomew, M. J., Hardin, J., Feng, Z., Thalman, R., and Machado, L. A. T.: The Green Ocean: precipitation insights from the GoAmazon2014/5 experiment, Atmos. Chem. Phys., 18, 9121-9145, https://doi.org/10.5194/acp-18-9121-2018, 2018.

**21 - Pg 28, ln 579: Strictly speaking, aggregates are not formed through vapor diffusion but through aggregation of more pristine crystal types that are grown from vapor diffusion. There may be a second issue here in that the aggregates category represents more than just the process of aggregation?**
We agree! Ice crystals grow by vapor diffusion, whereas aggregates are formed through aggregation processes of pristine ice. The text was modified accordingly and now reads:
*"In this cloud region, ice crystals grow by vapor diffusion until to have a sufficient weight to start falling and forming aggregates (Houze, 1997)."*

**22 - Pg 29, ln 598: This sentence confuses me. Perhaps you mean "the production of larger raindrops results mostly from ice microphysical processes?"**
Corrected as suggested.

**23 - Pg 30, ln 615: "the higher the aerosol concentration is, the more the coalescence process is suppressed (thus, leading to smaller particles)." Are you talking about in the warm (rain) phase here? I thought this discussion was related to aggregates and ice crystals. Additionally, this is only half the story. In the warm phase, coalescence may be suppressed with increased aerosol, but if updrafts are strong (as they tend to be in the dry season as stated in 611), more water may be pushed above the melting layer resulting in more mixed phase processes such as graupel production… but what is the effect on ice crystals and aggregates?**
We agree it was not clear. Therefore the section about the potential impact of aerosol concentrations has been modified as follows:

*"In terms of aerosol concentrations, the wet Amazonian season is known to be much cleaner than the dry season (Artaxo et al. 2002). With this regard, Williams et al (2002), Cecchini et al (2016), or even Braga et al (2017) highlighted its impact on the microphysical development of tropical cloud particles, showing that high aerosol concentrations may lead to smaller liquid particles within strong updraft regions. Well, small drops are known to freeze at colder temperatures by inhibiting the ice multiplication processes (Hallet and Mossop, 1974), and may account for the wet/dry season differences observed."*

Braga, R. C., Rosenfeld, D., Weigel, R., Jurkat, T., Andreae, M. O., Wendisch, M., Pöschl, U., Voigt, C., Mahnke, C., Borrmann, S., Albrecht, R. I., Molleker, S., Vila, D. A., Machado, L. A. T., and Grulich, L.: Further evidence for CCN aerosol concentrations determining the height of warm rain and ice initiation in convective clouds over the Amazon basin, Atmos. Chem. Phys., 17, 14433-14456, https://doi.org/10.5194/acp-17-14433-2017, 2017.

Hallett, J. and Mossop, S. C. C.: Production of secondary ice particles during the riming process, Nature, 249, 26–28, 1974.

**24 - Table 2: Please define S, V, T, and n.**
Done ! Please now reads:
*"Table 2: Distance formulas for the weighted, centroid and Ward linkage rules. Here, S and T are two clusters joined into a new cluster, whereas V is any another cluster. $n_S$, $n_T$, $n_V$ are the number of objects contained in the clusters S, T, V, respectively."*

**25 - Fig. 1: Might I suggest adding the current study on the diagram to make it clear to the reader how it fits in to previous work?**
Done.

**26 - Fig. 3: Is it surprising that 1 cluster explains 88% of the variance for most of the methods?**
Not that much. The dataset used in the study has been cleaned from any potential artefacts, ground clutters, … and from this point of view the dataset is quite homogeneous. Note that, by starting from raw data, the variance explained would have been lower.

**27 - Fig. 4: The green and blue of 7S and 6S in panel (g) are nearly indistinguishable in my printed version.**
We fully understand. Unfortunately, you have to notice that it is difficult to define a readable colorbar for each linkage rule outputs with 12-13-14 clusters… The panel (g) refers to Ward linkage rule, which has not been selected to deal with latter in the document. That is why we decided to do not change the colorbar of the panel (g).

**28 - Fig 6: I was somewhat surprised to see "liquid" up to 8 km. However, I looked up temperatures for the region that might be associated with 8 km, and found them to be only -10 to -15 C. 1) is that correct, and 2) perhaps a temperature reference for 8 km could be added in the text (e.g. ln 355) for reference for the reader?**
Good idea. Corrected!

**Responses to Reviewer #2**

We thank you for the thoughtful comments and changes suggested in your review of our manuscript. Our point-to-point responses are developed hereafter, along with an indication of changes made in the revised version of the text.

**1 - The authors have rephrased the description of using the melting layer as a parameter to detect liquid-ice delineation (lines 146-147 in the revised submission). But the content after this has not been revised accordingly (lines 147-151), which is still misleading.**
Corrected as suggested.

**2 - The response about calibration of ZDR should be included in the main text as well. In addition, the authors stated that the attenuation correction would not significantly change the results, which is very subjective. The authors referred to Dolan and Rutledge (2009) regarding this point. But Dolan and Rutledge (2009) was devised from a very different aspect (i.e., scattering simulation and fuzzy logic application, no clustering).**
We agree. More technical aspects about calibration of $Z_{DR}$ has been included in section 2. Please now reads:
*"The calibration of $Z_{DR}$ has been adjusted by using vertically pointing scans for cases with no rain attenuation (drizzle/light rain). This method allows to temporally calculate the $Z_{DR}$ offset since 0 dB is expected. The offset has been then removed in subsequent $Z_{DR}$ measurements. A second analysis of $Z_{DR}$ was occasionally realized by checking $Z_{DR}$ values within stratiform light rain medium and characterized by $Z_H$ values between 20 and 22 dBZ. The expected $Z_{DR}$ value was 0.2 dB as showed by Illingworth and Blackman 2002 or Segond et al. 2007."*

Also, potential overcorrection of $Z_{DR}$ due to the ZPHI method has been also mentioned in the discussion of the revised manuscript.
*"These discrepancies might be attributed either to an inaccurate attenuation correction or inherent tropical characteristics involved within microphysical ice growth. Although we considered a limited radar coverage, regions with high SNR values, as well as only precipitation events having a dry radome, the ZPHI method may still lead to overcorrection, especially on $Z_{DR}$ in strong convective cases when the Mie-scattering may dominate the precipitation regions. Another explanation of these discrepancies may also rely on tropical atmospheric characteristics that present higher tropospheric humidity profiles together with higher incident solar radiation, playing an important role in comparison to mid-latitudes."*

Finally, we would like to mention that we referred to Dolan et al (2009) in terms of clusters' contents and not attenuation correction. Although they used a different HCA technique, our results agree well with their findings (as well as those of Grazioli et al, 2015).